# Structural Performance of Thin-Walled Twisted Box-Section Structure

**Shijun Wang [1,2], Zan Wang [2], Chang Ping [1], Xing Wang [1], Huiying Wu [2], Jian Feng [2] and Jianguo Cai [2,\*]**

1   Economy & Technology Research Institute, Gansu Electric Power Corporation, State Grid, Lanzhou 730000, China; angelofkill@126.com (S.W.); pc519501410@foxmail.com (C.P.); wx19930919@126.com (X.W.)

2   School of Civil Engineering, Southeast University, Nanjing 210096, China; wangz174255@126.com (Z.W.); 220201206@seu.edu.cn (H.W.); fengjian@seu.edu.cn (J.F.)

\*   Correspondence: j.cai@seu.edu.cn

**Abstract:** The light weight and high strength-to-mass ratio of thin-walled boxed sections have incited interest in their widespread use in the construction of domes. However, the installation of these sections in forming the dome geometry has induced initial twists and curving features, to which their mechanical response has rarely been explored. Therefore, the structural performance of a structure with thin-walled twisted box sections is numerically studied in this paper, employing ANSYS, the verification of which is carried out through a comparison with experimental results. Additional components examined include the longitudinal stiffening rib, diaphragm, and web. The effects of variations in the thicknesses of these member plates on the mechanical behaviors are investigated. In general, the ultimate capacity of the structure is improved by increasing the thickness of the longitudinal stiffening rib, diaphragm, and web, but the strengthening effect of the stiffener is limited by a certain thickness enhancement. The common failure mode of the initial model is found to be an overall elastic-plastic buckling. A reduction in the thickness of the stiffener or web creates a curving deformation zone in the lower arch at the ultimate capacity, whereas the diaphragm thickness has little effect on the failure mode of the model.

**Keywords:** twist-curved; thin-walled; dome; finite element; ultimate capacity

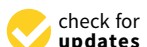



## 1. Introduction

With recent developments, thin-walled box-section structures have been extensively used in engineering components thanks to their impressive performance, such as a high stiffness-to-weight ratio and high torsional and bending rigidities [1–3]. For example, box-section girders [4–6] and columns [7] are familiar applications of civil engineering structures. Because the three-dimensional behavior of the thin-walled box-section structure contains torsion, distortion, and bending in longitudinal and transverse directions, many research works have focused on the practical optimal design and analysis of thin-walled box-section structures [8–10]. Square sections with various internal topologies are commonly used in practice: a solid section, a foam-filled tube with monolithic walls, a hollow tube with walls made from sandwich plates, and a hollow tube with walls reinforced by internal stiffeners [11]. Depending on the design and requirements, thin-walled box-section structures can be constructed in different box girder configurations, including straight box-section structures [12] and curved box-section structures [13,14]. Due to their high torsional rigidity, they are well suited for use in curved-shaped box-section structures.

If they have a light curvature, curved structures can be treated as straight ones [15]. However, this method, adopted to simplify the analysis and design procedure, has limitations. Because high-capacity computational systems are available, it is possible to make higher-level investigations. The influence of shear deformability on the mechanics of the thin-walled composite curved beams with open and closed cross-sections was illustrated by Piovan and Cortinez [16]. Peres et al. [17] presented the first-order GBT formulation for

naturally curved thin-walled members with a constant ending curvature. It was noted that due to the curved cross-section, shell-like buckling is a potential mode of failure. Jobbágy and Ádány [18] extended their examination of buckling behavior and concluded that such shell-like behavior is critical only in the case of unusual cross-section geometries. Peres et al. [19] proposed a mixed finite element method based on GBT to analyze the first-order behavior of naturally curved beams with the circular axis (without pre-twist) with deformable thin-walled cross-sections. Kim et al. [20,21] proposed an improved displacement field formulation and the stiffness matrix for the spatially coupled stability analysis of a thin-walled curved beam with non-symmetric cross-section subjected to uniform compression. Its performance has been analyzed with the ANSYS shell elements. Razaqpur and Li [22] proposed a new curved thin-walled multicell box girder element, extending the computational ability to handle extension, flexure, torsion, torsional warping, distortion, distortional warping, and shear lag effects based on both Vlasov's thin-walled beam theory and the finite element technique. Song et al. [23] presented a novel finite element analysis of curved concrete box girders. It has been shown that shell and beam elements can be efficiently used to reduce the number of elements, while maintaining the accuracy of the model.

Even though much research has been carried out in the restricted area of the analytic simplification of thin-walled box-section structures, predicting the behavior of the components imposes a certain degree of challenge due to the extra torsional effects resulting from their geometric configuration. In some early studies of beam element behaviors, Dabrowski [13] proposed the use of the distortion angle to indicate the magnitude of the cross-sectional distortion. Bazant and Elnimeir [24] introduced the skew-ended finite element, permitting the implementation of the theory of non-uniform torsion. Yu et al. [25] presented an asymptotically correct theory for initially twisted thin-walled composite beams. They implemented a variational asymptotic method, applicable to all types of thin-walled beams, including strips and closed or open sections. Their method comes without the limitation of using a separate theory for different beam types, as required by traditional techniques. Zhou et al. [26] investigated the effects of the pre-twisted angle ratio on the structural behavior of steel box-section columns. Based on the concept of effective torsional stiffness, Shen et al. [27] proposed a new method for calculating the post-cracking twist angles in girders to predict maximum deflections in curved concrete. Arici and Granata [28] analyzed the straight and curved thin-walled structures on elastic foundations using the Hamiltonian structural analysis method and proposed a unified comprehensive theory for closed and open thin-walled cross-sections, as well as compact sections, by considering non-uniform torsion and distortion plus torsional moment deformation effects.

Due to valuable features such as the light weight and high strength of thin-walled box-section structures, they have been broadly employed in the construction of domes. The majority of classical analytical solutions are available to analyze the structural bearing capacity and deformation modes of a hollow tube with walls. However, contemporary box structures may differ in their shape and materials. Considering cross-sectional designs, the geometric and material parameters of thin-walled box-section structures require further study. Taking into account geometrical non-linearity, classical solutions lose their efficacy in the prediction of the mechanical properties of elastomers. Most of these problems are solved using semi-empirical and semi-theoretical formulas or finite element methods. Although scientific efforts have been focused on specific irregular thin-walled structures, yielding various efficient calculation methods, they are not applicable to arbitrary structures or especially to those of complex types.

In forming the dome's outer geometry, some degrees of twist and curviness have been introduced to thin-walled box-section members. The analysis of thin-walled box-section members with a composition of twists and bends, however, remains relatively rare. Due to the influences of the spatial distortion of the components, coupled with the complex stress formation in the existence of tension, bending, and shearing, it can be too subtle for ordinary beam elements to stimulate and capture the effects accurately. Moreover, the flow

of force in the member at the joint zone is inherently complicated, whereas the influence of stiffeners may also be considered as an additional factor. To address the above-mentioned limitations of existing beam elements, the application of shell elements is proposed in the current study. The structural spatial deformation and complex stresses in the presence of tension, bending, and shear, coupled with the influences of the addition of a stiffener, diaphragm, and web in the structural configuration, introduce further challenges to the analysis process. Hence, the structure can only be effectively modeled using shell elements.

This paper aims to study the structural performance of thin-walled twisted box-section structures, focusing particularly on the ultimate capacity of the structure. The effect of geometric nonlinearity is considered, and the effect of material nonlinearity is also considered in the analysis of the ultimate bearing capacity. The examined parameters include the thickness effects of the member plates, including those from the longitudinal stiffening rib, diaphragm, and web in influencing the stress distribution, as well as the corresponding deformation mode. Conclusions and major findings are then proposed at the end of the paper.

## 2. Dome Model

The considered dome structure has been taken from the museum of Shandong, China, as shown in Figure 1. The dome adopts thin-walled twisted box sections as its members. The general form of the dome resembles a semi-ellipsoid, spanning 63 m with a height of 21.5 m. In detail, the dome is made up of an outer arch (a four-layered arch structure composed of thin-walled twisted box-section members), an inner reticulated shell, middle braces, and beam string structures at the top with a total steel mass of about 1300 tons. Four stacks of curving and twisted arches form the skeletal configuration of the dome with a general geometrical central outline as described by the ellipsoid equation. The equation of a general geometrical central outline is as follows:

$$X^2/31.5^2 + Y^2/31.5^2 + Z^2/21.5^2 = 1 \text{ (units in m)} \tag{1}$$

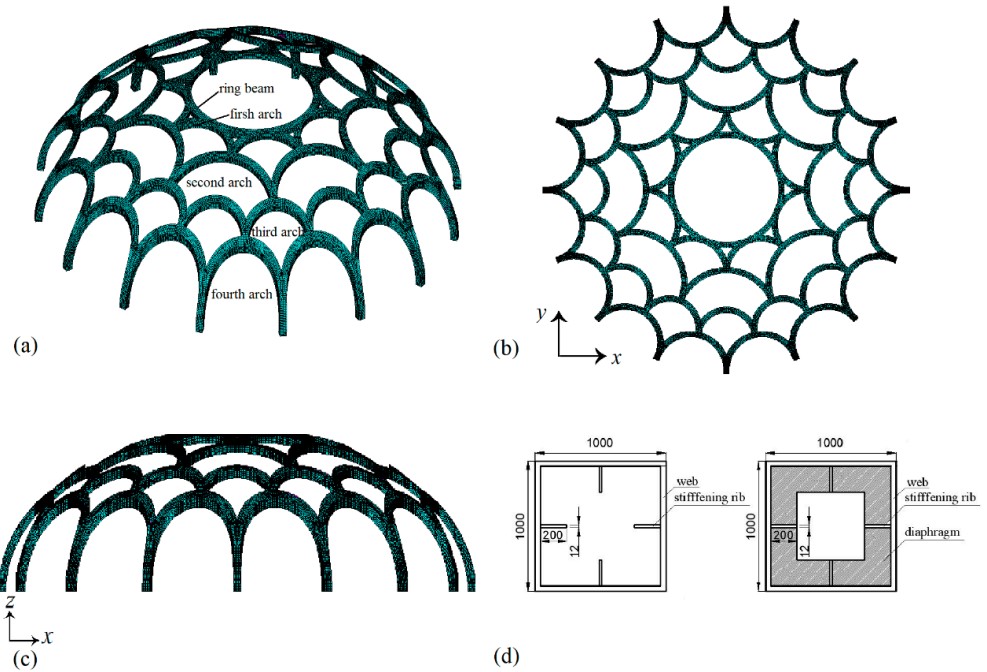

**Figure 1.** The dome with the stacking of the arches, (**a**) classification of arch layers, (**b**) top view of the dome, (**c**) lateral view of the dome, (**d**) cross section of twisted box section beams with stiffening ribs and cross section with stiffening ribs and diaphragms (units in mm).

The stiffening ribs are arranged vertically in the middle of each web along the length direction of the webs. Its specific dimensions are shown in Figure 1d. The horizontal diaphragms with square holes are arranged along the length of the web, with a width of 200 mm and a thickness of 16 mm. From the first layer to the fourth layer, the spacing of the diaphragm is 2.4 m, 3.2 m, 2.4 m, and 4.8 m, respectively. The geometrical and material details of each arch layer are summarized in Table 1.

**Table 1.** Geometrical and material details of each arch layer.

| Structure | Height (mm) | Arch Curvature $(mm^{-1})$ | Central Angle (Corresponds to Arch Curvature) | Thin-Walled Twisted Box Section Dimension (mm) | Material |
|---|---|---|---|---|---|
| Ring beam | - | $1.043 \times 10^{-4}$ | 360° | $1000 \times 1000 \times 16$ | |
| First arch layer | 1500 | $1.675 \times 10^{-4}$ | 120° | $1000 \times 1000 \times 16$ | |
| Second arch layer | 3200 | $1.163 \times 10^{-4}$ | 130° | $1000 \times 1000 \times 16$ | Q235 |
| Third arch layer | 3200 | $1.965 \times 10^{-4}$ | 150° | $1000 \times 1000 \times 16$ | |
| Fourth arch layer | 13,600 | $1.623 \times 10^{-4}$ | 170° | $1000 \times 1000 \times 20$ | |

The following load case is considered: prestress + 1.485 dead load + 1.078 live load, as the dead load and live load are all taken as 1.0 kN/m². The displacement and stress distributions of the web, longitudinal stiffening rib, and diaphragm under a single applied load magnitude (1*P*) are shown in Figure 2. The maximum displacement of the structure under 1*P* is 22.53 mm. The maximum stress is 83 MPa at the bottom arch foot. The first- and second-order deformation modes experienced by the dome are translational movements in the horizontal *X* and *Y* directions, whereas the third-order mode is the torsional vibration. The ultimate load that the structure can withstand is 9.41*P*, with a maximum displacement of 322.8 mm.

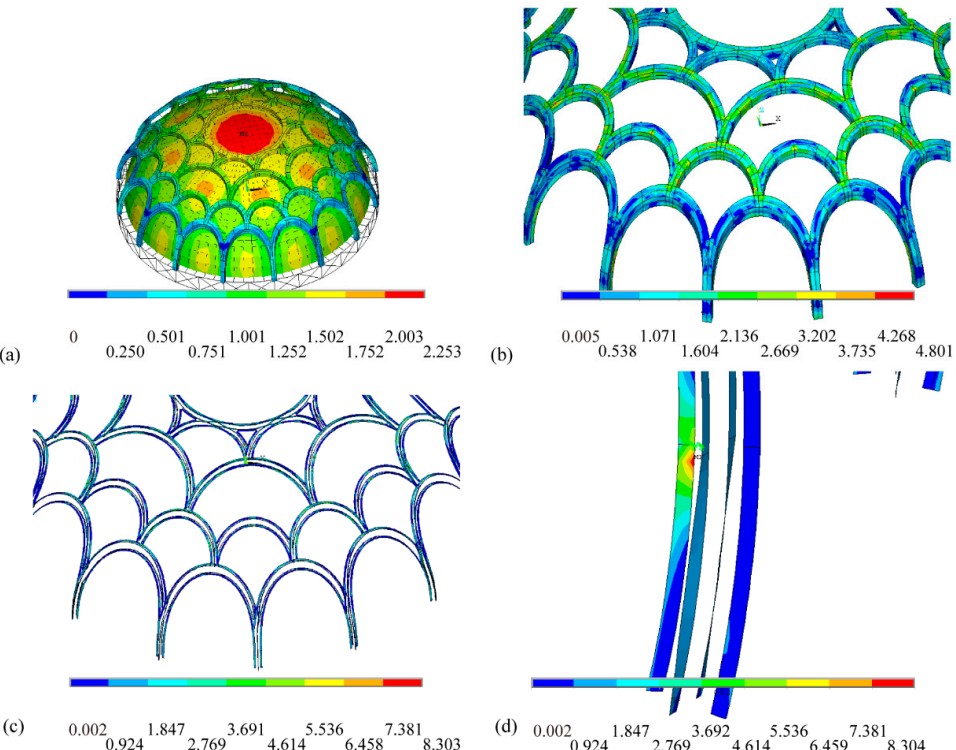

**Figure 2.** (**a**) Displacement and stress distributions of (**b**) web, (**c**) longitudinal stiffening rib, and (**d**) diaphragm under applied 1*P* load.

### 3. Finite Element Analysis of the Twisted Box-Section Model

Since the dome is symmetrical with a repetitive pattern of similar structural units, a structure with a common juncture node connecting the second arch and the third arch levels formed by four twisted thin-walled box members is considered for detailed analysis. According to the initial analysis, the stress generated by the axial force in the members is relatively large. To actively adjust the axial force of the structure, due to the inconvenience of applying constraints on the upper part (for the experimental study described below), the upper end of the second arch layer has been modified to form a cantilever.

#### 3.1. Finite Element Model

For the analysis, a representative finite element model comprising thin-walled twisted box-section members was created in ANSYS [29], as shown in Figure 3a. The FEA model consists of a common node connecting the second arch and the third arch layers, and it is connected to the joint by four twisted thin-walled box members. The specific dimensions of the specimen are shown in Figure 4. The web thickness of the model is 6 mm. For consistency, all components of the model use Q235 steel. In addition, the longitudinal stiffening ribs, with a width of 75 mm and a thickness of 4 mm, are arranged vertically in the middle of each web along the length direction of the curved beam webs. The horizontal diaphragms with square holes are arranged along the length of the web, with a width of 75 mm and a thickness of 6 mm. The spacing of diaphragms arranged in the second and third arch layers is 1200 mm and 900 mm, respectively. At the joint of the second and third arch layer, the diaphragms are arranged compactly with a distance of 350 mm.

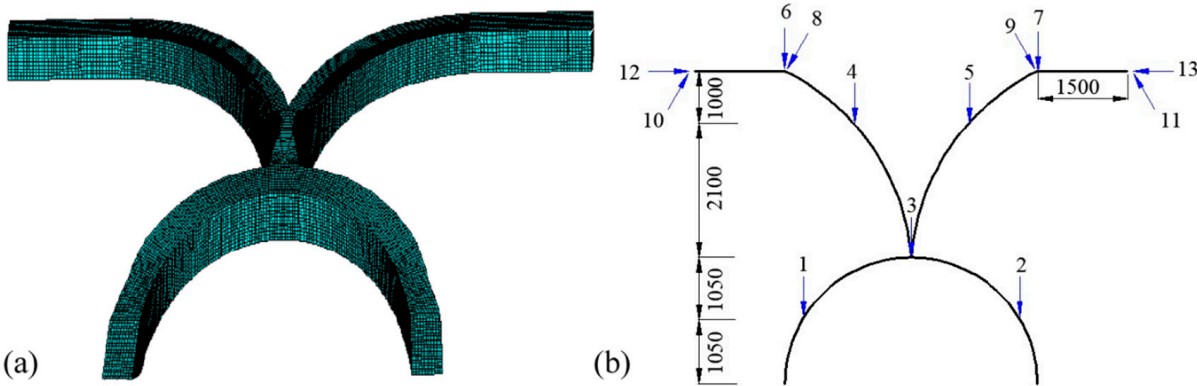

**Figure 3.** (**a**) Finite element model for the initial model and its corresponding (**b**) loading points and loading directions (units in mm).

The model utilizes the SHELL181 element to simulate the twisted box member (including internal stiffening ribs and diaphragms). The SHELL181 element is a four-node element with six degrees of freedom per node, thus being suitable for the nonlinear analysis of thin and medium-thick shells. Considering the accuracy and cost-of-calculation results, the mesh size needs to be accurately defined. After the verification of convergence, the meshed model converged with node and element numbers of 46,038 and 2120, respectively. The bottom of the model is fixed. In order to make the stress of the local model consistent with the stress of the global model under loads, the latter is continuously adjusted throughout the model analysis. The adopted material properties are as follows. For the web and diaphragm, the yield strength is 322.2 MPa, the elastic modulus is 206,000 MPa, and the Poisson ratio is 0.3. For the longitudinal stiffening ribs, the yield strength is 313.6 MPa, the elastic modulus is 201,000 MPa, and the Poisson ratio is 0.3. The loading points and the corresponding magnitude on each point are displayed in Figure 4b and summarized in Table 2. Loading points 8 and 9 are vertical to the arch, loading point 10 is parallel to loading point 8, and loading point 11 is parallel to loading point 9.

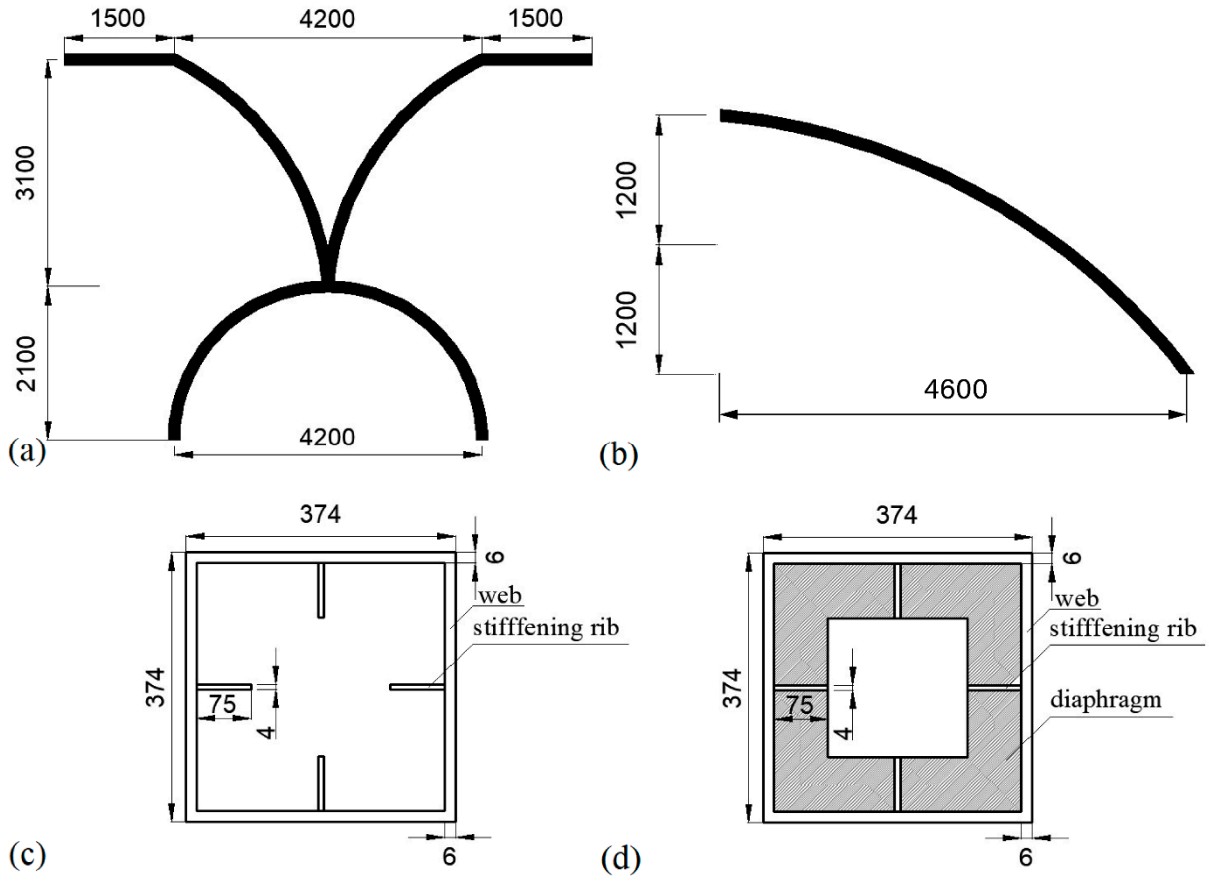

**Figure 4.** Specific geometrical details of the FEA model—(**a**) whole model, (**b**) lateral view of the model (relative position in the dome), (**c**) cross-section with stiffening ribs, (**d**) cross section with stiffening ribs and diaphragms (units in mm).

**Table 2.** Load magnitude associated with the loading point shown in Figure 3b.

| Loading Point | Load Magnitude, $P$ (kN) |
|---|---|
| 1 | 11.25 |
| 2 | 11.25 |
| 3 | 11.25 |
| 4 | 42.8 |
| 5 | 42.8 |
| 6 | 4 |
| 7 | 4 |
| 8 | 128.7 |
| 9 | 128.7 |
| 10 | 34.51 |
| 11 | 34.51 |
| 12 | 62.33 |
| 13 | 62.33 |

*3.2. Numerical Results*

The wall at the juncture node zone enters the plastic regime first at 2.8*P*, whereas the diaphragm and longitudinal stiffening ribs in the same zone reach plasticity at 3.2*P*. Plasticity is experienced by the diaphragm and the longitudinal stiffening rib in the lower part of the lower arch at 3.4*P* and 3.8*P*, respectively. The increase in the applied plastic zones in these two locations has brought the whole model to achieve the ultimate capacity at 4.744*P*. Upon close inspection, there is no visible curving and plastic strain in the longitudinal stiffening rib before the model fails. When the model reaches its ultimate

capacity, the maximal plastic strain in the web, longitudinal stiffening rib, and diaphragm are as listed in Table 3.

**Table 3.** Maximal plastic strain in each location.

| Component | Location | Maximal Plastic Strain |
|---|---|---|
| Web | Node zone | $1.6 \times 10^{-2}$ |
| | Middle and lower part of the lower arch | $4.9 \times 10^{-3}$ |
| Longitudinal stiffening rib | Node zone | $4.0 \times 10^{-3}$ |
| | Middle and lower part of the lower arch | $3.5 \times 10^{-3}$ |
| Diaphragm | Node zone | $4.46 \times 10^{-3}$ |
| | Middle and lower part of the lower arch | $2.25 \times 10^{-3}$ |
| Web | Node zone | $1.6 \times 10^{-2}$ |

The contour diagrams of the displacement, plastic Mises stress, and plastic strain in the top and bottom surfaces of the web in the ultimate-capacity case are shown in Figure 5. The stress and plastic strain in the diaphragm and longitudinal stiffening rib are shown in Figure 6 (a displacement magnification factor of five has been used to enhance the legibility of the images).

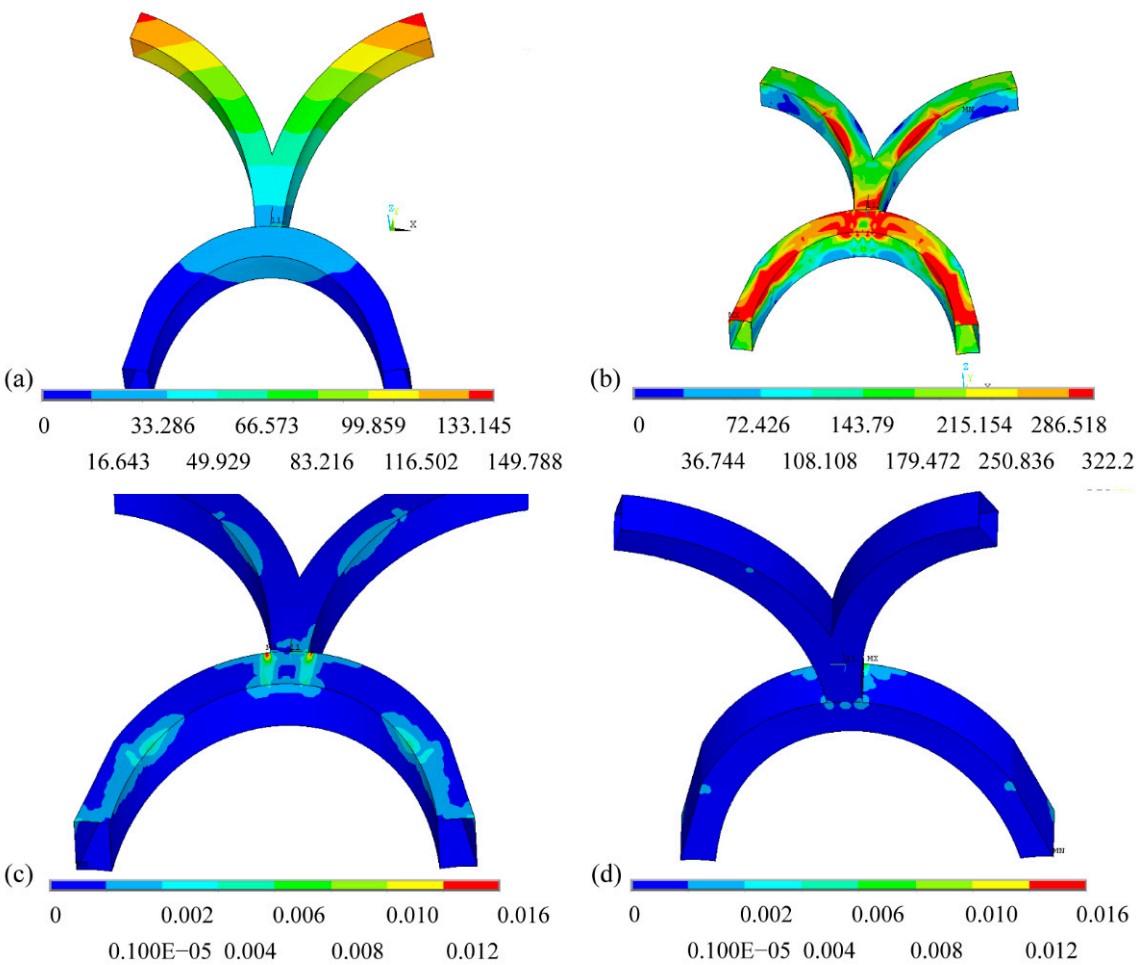

**Figure 5.** (**a**) Displacement, (**b**) Mises stress, and plastic strain in the (**c**) top and (**d**) bottom parts of the web for the initial model.

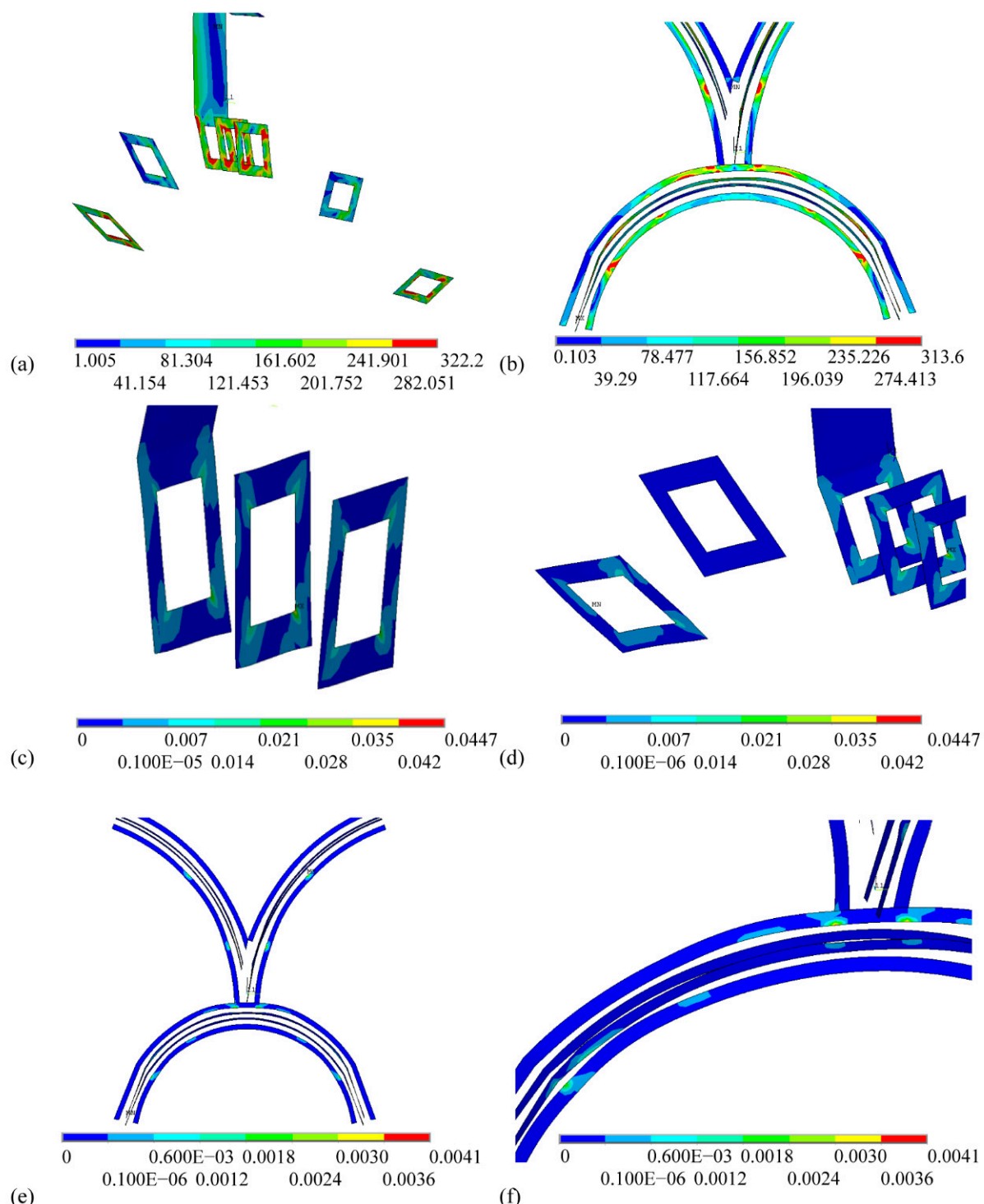

**Figure 6.** Mises stress in the (**a**) diaphragm and (**b**) longitudinal stiffening rib and the plastic strain in the (**c**) diaphragm in the juncture node zone, (**d**) lower diaphragm, (**e**) longitudinal stiffening rib, and (**f**) magnified perspective of the longitudinal stiffening rib for the initial model.

There is no visible curving deformation at the juncture node, as shown in Figure 6a. In Figure 6c,d, the plastic zones of the web are concentrated in the node zone and lower arch. Again, the bending deformation is absent in the diaphragm and longitudinal stiffening rib, which have reached the plastic stage before the occurrence of the failure in the model. The plasticity development in the diaphragm in the node zone is more obvious with a high concentration at the corners, which is marked in Figure 6c.

### 3.3. Comparison of Experimental and Numerical Results

A laboratory specimen was fabricated to verify the finite element results, as shown in Figure 7.

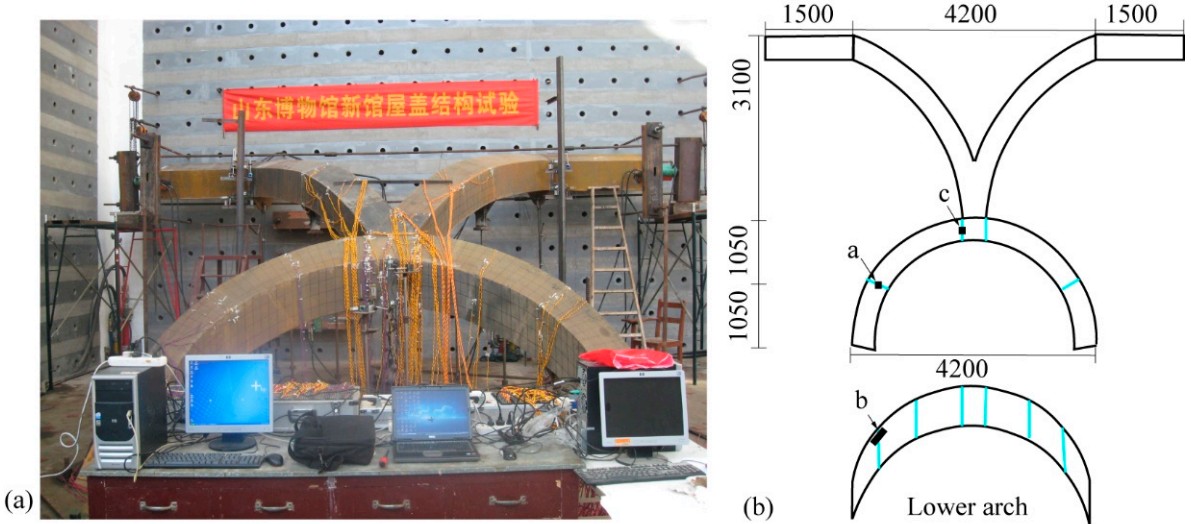

**Figure 7.** (**a**) The specimen and (**b**) its geometrical details (units in mm).

Representative points at the *a* and *b* sections on the lower arch and *c* section have been selected for strain comparison, as shown in Figure 8a–d, respectively. The *a*, *b*, and *c* sections are marked in Figure 7b. Displacement plots of the juncture node in the horizontal (parallel to the wall) and vertical directions (perpendicular to the wall) are shown in Figure 8e,f, respectively.

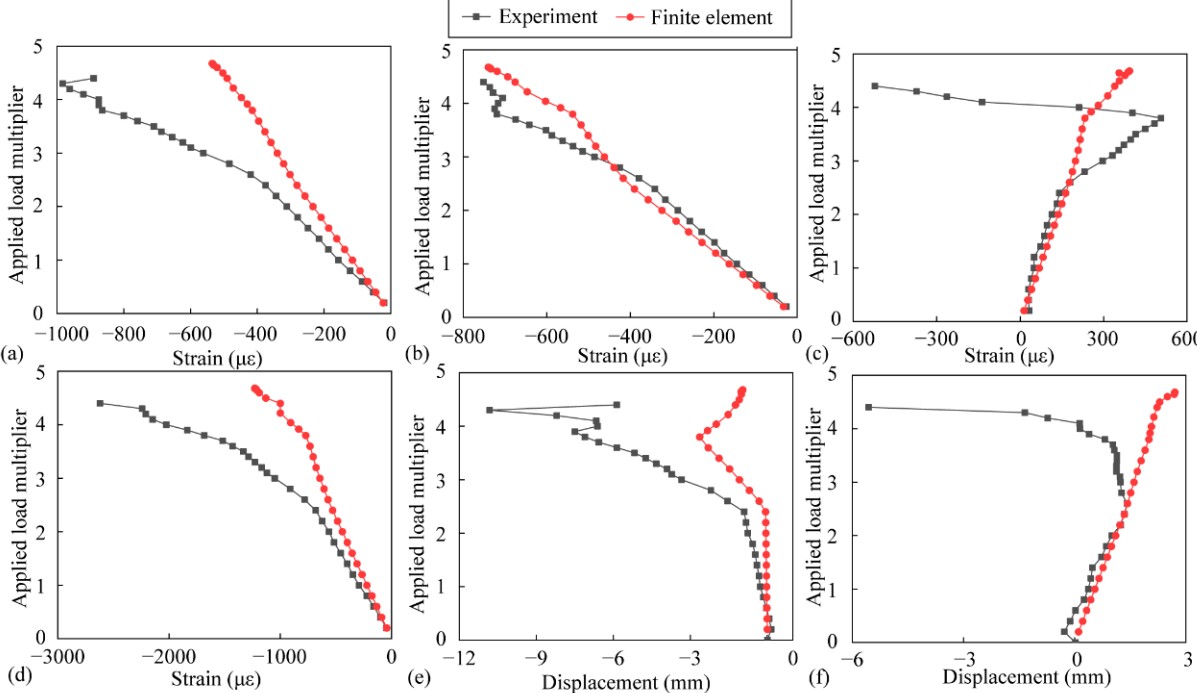

**Figure 8.** Comparative plots of strains at the (**a**) a and (**b**) b sections, (**c**) first principal strain and (**d**) principal strain at the c section, and displacement of juncture node in the (**e**) horizontal and (**f**) vertical directions.

It can be observed that the differences between the finite element and test results of most locations before 2.4$P$ were very small, whereas some regions showed deviations after that. The differences were mainly determined by the sliding in the foot of the specimen. In the experiment, when the load reached 2.4 times the load case magnitude, the foot at the left arch slipped in the radial direction, and at 3.8 times, the foot at the right arch slipped in the radial direction.

## 4. Parametric Analysis

In order to investigate the mechanical behaviors of the model, the effect of the thicknesses of the longitudinal stiffening rib, diaphragm, and web were considered. The out-of-plane stability of the web was improved via the installation of a longitudinal stiffening rib. As a result, the ultimate bearing capacity of the structure was also enhanced. The diaphragm worked as the lateral support for the longitudinal stiffening ribs, reducing the span of the longitudinal stiffening ribs and the free length in case of instability under pressure. When the wall thickness was small, the damage pattern of the structure included elastic-plastic overall instability and local instability. In this paper, the thickness of the initial model was $T_{ls}$ = 4 mm, $T_d$ = 6 mm, and $T_w$ = 6 mm. For the parametric analysis, 0, 2 and 6 mm were selected as $T_{ls}$ for the performance comparison with the initial model of thickness $T_{ls}$ = 4 mm. Four and eight millimeters were selected as $T_d$ values for performance comparisons with the initial model of thickness $T_w$ = 6 mm. Four and eight millimeters were selected as $T_w$ values for performance comparisons with the initial model of thickness $T_w$ = 6 mm.

Realizing these effects, their mechanical behaviors were then studied in terms of deformation, stress, strain, ultimate capacity, and the failure mode of the model.

### 4.1. Thickness of Longitudinal Stiffening Rib, $T_{ls}$

Three thicknesses, $T_{ls}$ = 0, 2, and 6 mm except the initial one, were proposed to study the effects of the longitudinal stiffening rib on the ultimate capacity. The plastic strain in the web and its amplified perspective are shown in Figure 9a,b. When the longitudinal stiffening rib was omitted from the model, the web in the juncture node zone reached the plastic stage at 2.6$P$, whereas the middle and lower parts of the lower arch reached this state at 3.0$P$. Bending deformation was not visible in the model when the load level reached 3.6$P$. Differently, when the load attained 3.8$P$, the middle and lower parts of the lower arch bend rapidly with a fast increment in plastic strain. The model reached its ultimate capacity at 3.925$P$. There was a noticeable bending in the local part of the lower arch when the model reached its ultimate capacity.

The corresponding plastic strain in the web and its amplified view are shown in Figure 9c,d. When $T_{ls}$ = 2 mm, the longitudinal stiffening rib in the middle and lower parts of the lower arch were the first parts to achieve the plastic stage at 2.0$P$. Then, the longitudinal stiffening rib in the juncture node zone reached the plastic stage at 2.2$P$. The web and diaphragm in the node zone attained their plastic behaviors at 2.6$P$ and 3.2$P$, respectively. Subsequently, the same members in the middle and lower parts of the lower arch reached this state at 3.4$P$. When the load level was raised to 4.0$P$, there was an evident bending in the longitudinal stiffening rib but no such occurrence in the web. The ultimate capacity was achieved at 4.514$P$, with a gradual curving deformation in the web. The longitudinal stiffening rib was buckled locally before reaching the model failure, as shown in Figure 9d. Hence, the development of plastic strain in this member is evident.

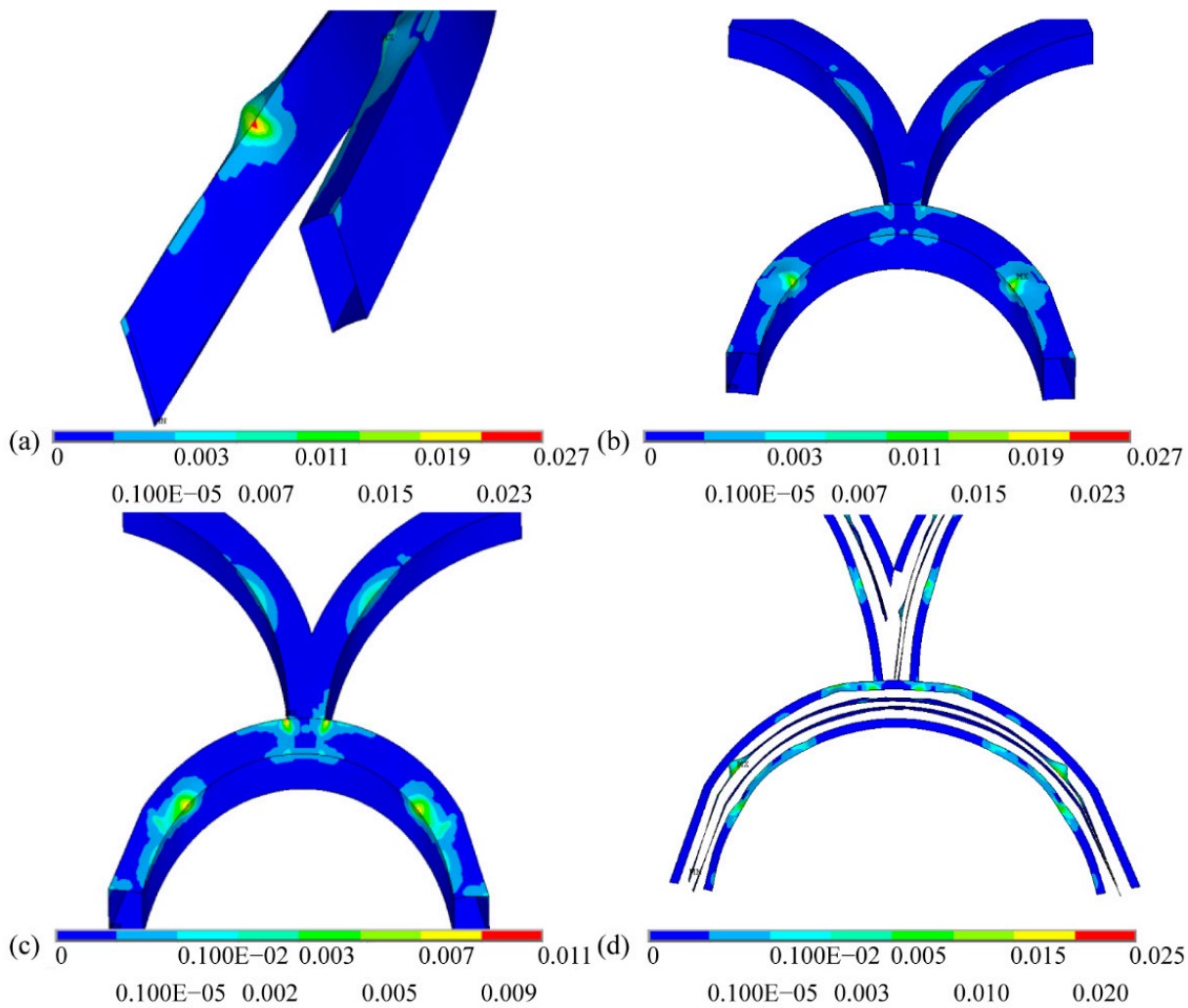

**Figure 9.** (**a**) Plastic strain and the (**b**) corresponding amplified view in the web at $T_{ls} = 0$ mm; plastic strain in the (**c**) web and (**d**) longitudinal stiffening rib at $T_{ls} = 2$ mm.

When $T_{ls} = 6$ mm, the web in the juncture zone reached its plastic stage at $2.8P$. The same members in the middle and lower parts of the lower arch achieved plasticity at $3.8P$. Although the two plastic zones continued expanding with the increasing of the load, the model attained its ultimate capacity at $4.826P$. The longitudinal stiffening rib entered the plastic stage at $4.3P$, later than other members in this model did.

According to the presented outcomes, considering four thicknesses, the effects of a longitudinal stiffening rib on the structure's behavior can be summarized into four aspects:

1.  Deformation

The load-displacement curves of several points are depicted in Figure 10. The points were selected from the top flange and front web of the juncture node zone and the curved zone.

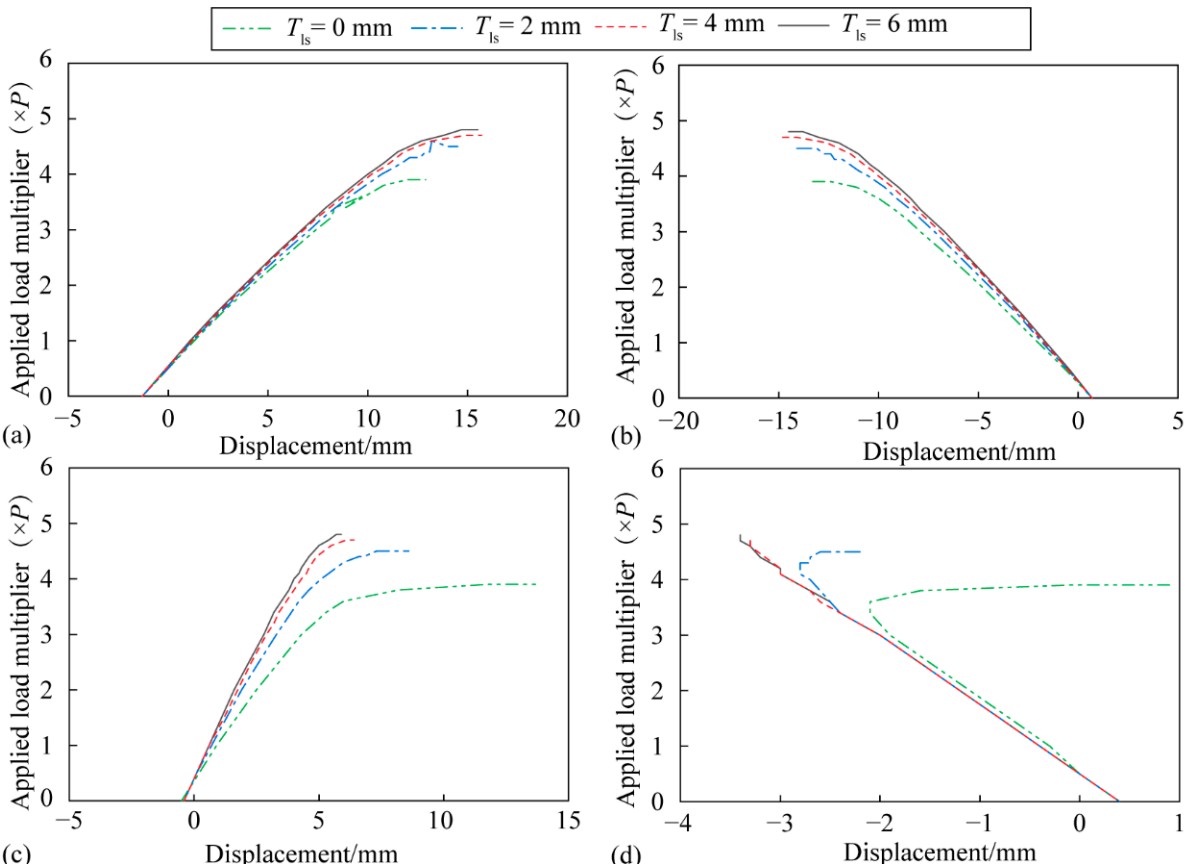

**Figure 10.** (**a**) Vertical displacement on the top flange, (**b**) horizontal displacement on the front web at the juncture node zone and (**c**) vertical displacement on the top flange, (**d**) horizontal displacement on the front web -in the curved location due to the change in the thickness of the longitudinal stiffening rib.

According to Figure 10a,b, the load-deformation relationships in the juncture node zone are close to each other, indicating that the effect of longitudinal stiffening rib on deformation can be ignored in the juncture node zone and that the whole rigidity of the model is less affected by the longitudinal stiffening rib. The deformation in the curved zone of the lower arch in each situation is close to the initial model before 3.4*P*. Under a high load and when the thickness is less than 4 mm, there are deformations (curved phenomenon) at this zone, i.e., the top flange curves upward and the outer web dents inward. As depicted in Figure 10c,d, the curves for 0 and 2 mm thicknesses deviate from the common behavior when the load level is close to the ultimate capacity. This illustrates that the longitudinal stiffening rib is key to the deformation in the curved zone.

2. Stress

According to the deformation analysis, there is a curving deformation in the lower arch when the thickness of the stiffener is less than 4 mm. To identify the reason for this, the axial, tangential, and shear stresses in the maximally curved part under 1*P* were analyzed, as shown in Figure 11. The horizontal axis represents the loading points in the top flange of the curved zone from left to right.

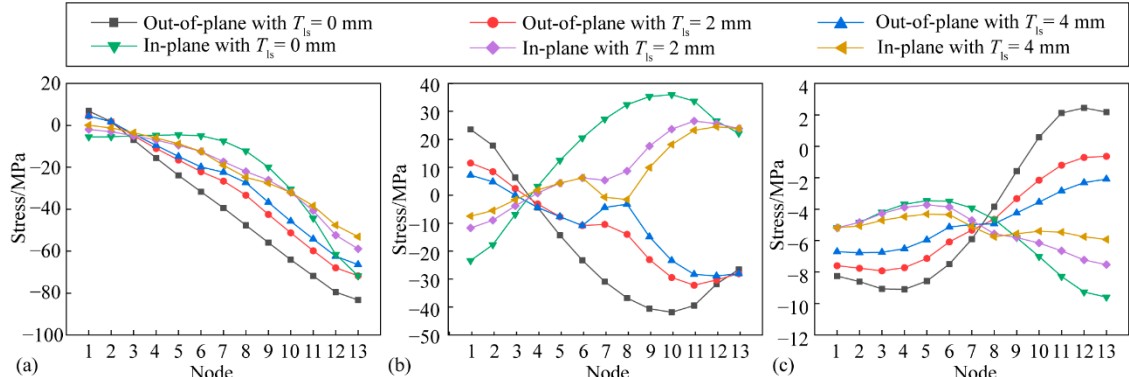

**Figure 11.** (**a**) Axial, (**b**) tangential, and (**c**) shear stresses in the top flange of the curved zone under 1*P* loading.

In Figure 11a, the axial stress in the cases of $T_{ls}$ = 2 mm and $T_{ls}$ = 4 mm are close to each other, which means that the M11 moment along the axis in the plane formed by the axis and normal of the member plate was small. When the stiffener was omitted, the axial stress in each point increased, and the maximal increment was 18.4 MPa.

In each situation, the in-plane stress was transformed to out-of-plane, which meant that the resulting stress was small when the in-plane moment M22 along the normal of member plate formed by the tangent and normal of member plate was larger. The tangential stress in the model with $T_{ls}$ = 4 mm was close to the one with $T_{ls}$ = 2 mm. When the stiffener was absent, the tangential stress increased visibly with the increasing of the difference between the in-plane and out-of-plane stresses. Meanwhile, the M22 moment in the plane formed by the tangent and normal of member plate increased visibly. This phenomenon illustrates that the longitudinal stiffening rib can be considered a middle brace to the member plate for reducing M22. As observed in Figure 11c, with the reduction of the stiffener thickness, the difference in shear stress between the inner and outer plates increases.

The axial and tangential strains in the curved zone under ultimate capacity are shown in Figure 12. In the initial model with 4 mm thickness, the difference between the axial strain in the outside and inside surfaces is smaller when the difference of tangential strain is larger. When the stiffener is omitted, the two differences are both larger because of the curved plate.

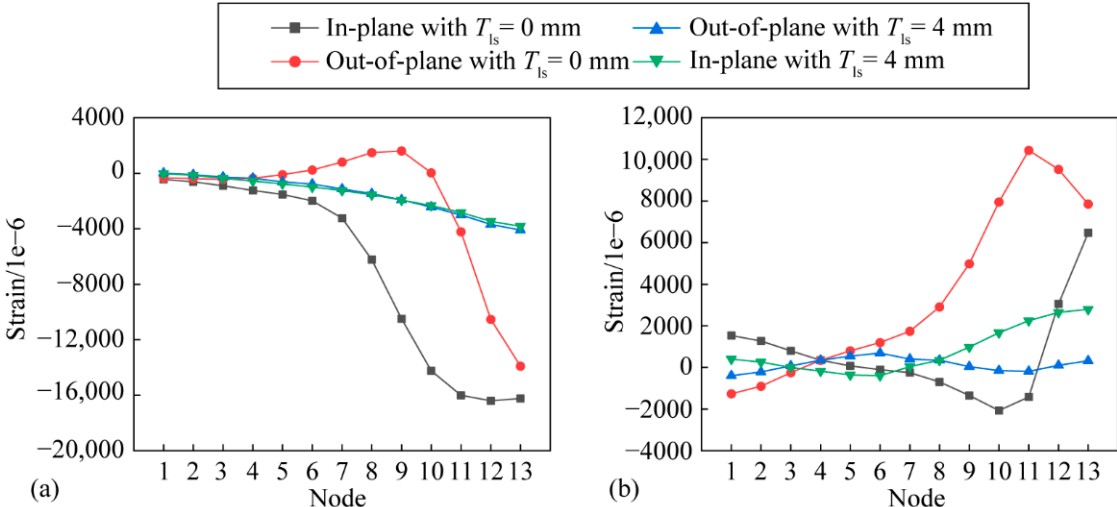

**Figure 12.** (**a**) Axial and (**b**) tangential strains in the top flange of the curved zone under ultimate capacity.

The moments M11 and M22 in the member plates with $T_{ls}$ = 4 mm are depicted in Figure 13. M11 and M22 in the longitudinal stiffening rib of the initial model ($T_{ls}$ = 4 mm) under 1*P* are both small. Nevertheless, when the stiffener is absent, the moments increase greatly. In addition, the maximal moment in the initial model under ultimate capacity is less than the maximal moment in the model with $T_{ls}$ = 0 mm under the same conditions.

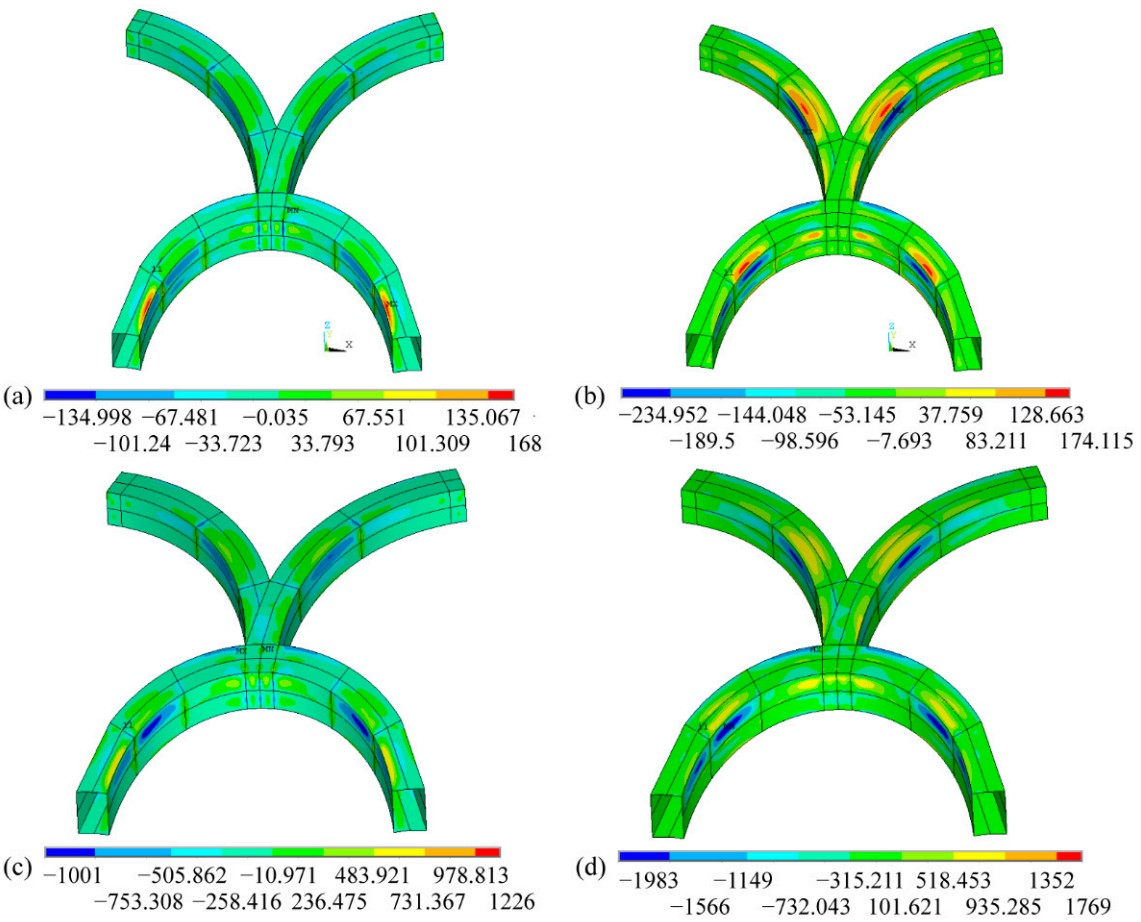

**Figure 13.** (**a**) M11 and (**b**) M22 under 1*P* and (**c**) M11 and (**d**) M22 under ultimate capacity when $T_{ls}$ = 4 mm.

3.  Failure mode

The failure modes in the proposed model can be described in three categories:

When the thickness of the stiffener is 4 mm, at the ultimate capacity, the plastic zones in the juncture node zone and the lower arch have developed fully. In this state, the slopes of load-displacement curves in the juncture node zone are all near to zero, thereby resulting in overall elastic-plastic buckling.

When the thickness of the stiffener is extremely small or negligible, the middle and lower parts of the model experience curving deformation at the ultimate capacity. In this case, the failure mode includes two instability modes: overall buckling and elastic-plastic local buckling.

When the thickness of the stiffener is in between the two above-mentioned cases, the failure mode of the model may include the local buckling depending on the value of thickness-at the ultimate capacity as the overall buckling mode always exists.

4.  Strain and ultimate capacity

The relationship between the ultimate capacity and the thickness of the longitudinal stiffening rib is shown in Figure 14. In addition, the maximal plastic strain in each part of the model under the ultimate capacity is shown in Figure 15.

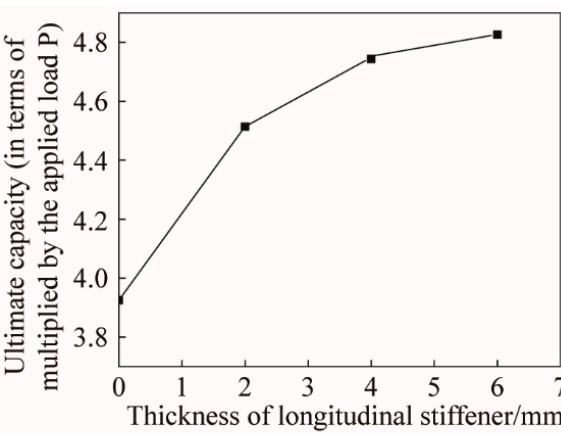

**Figure 14.** Relationship between the ultimate capacity and the thickness of the longitudinal stiffening rib.

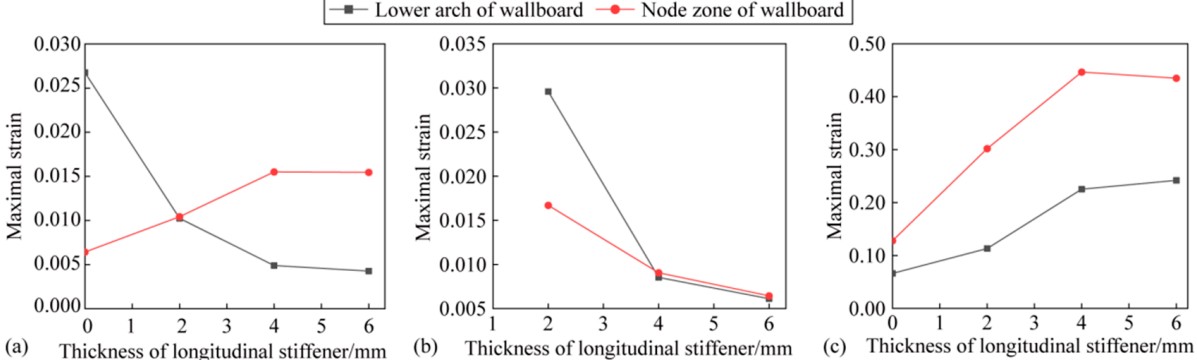

**Figure 15.** Maximal strain vs. thickness of the longitudinal stiffening rib in the juncture node zone and lower arch of (**a**) web, (**b**) longitudinal stiffening rib, and (**c**) diaphragm.

The ultimate capacity of the model increases with the thickness, as displayed in Figure 13. On the contrary, with an increase in thickness, the plastic strain in the web shows different tendencies in the juncture node zone and the lower arch, as shown in Figure 15a. In the node region, the plastic strain continues rising to a thickness of 4 mm. After that, the strain remains around 0.015. In the lower arch, the plastic strain drops sharply with the thickness up to 4 mm. A further increase causes only a small decrement in strain, down to about 0.005.

A decrease in thickness causes a sharp rise in the plastic strain in the juncture zone of the stiffener and an earlier buckling, as displayed in Figure 15b. During the loading sequence, the plastic strain in the lower arch also ascends but only gradually. When the thickness is greater than 4 mm, although the plastic strain variation is relatively smaller, its descending trend is clear, whereas those of the web and diaphragm remain unchanged.

The plastic strain in the diaphragm in the juncture zone and lower arch decreases quickly with the thickness, as shown in Figure 15c. This illustrates that the thinner the stiffener, the greater the concentration of plastic deformation on the curved parts. With a thickness of more than 4 mm, the changing tendency of the plastic strain in the diaphragm is close to stagnancy.

### 4.2. Thickness of Diaphragm, $T_d$

To examine the effects of the diaphragm on the mechanical behaviors, 4 and 8 mm were selected as $T_d$ values for performance comparisons with the initial model of thickness $T_d = 6$ mm.

It is worth noting that the plastic strain with $T_d = 8$ mm is less than that with $T_d = 4$ mm (not shown). When the diaphragm thickness of the initial model is changed from 6 mm to

4 mm, the webs in the juncture node zone reach the plastic phase at 2.6*P*. The other parts of the model arrive at plasticity successively: the diaphragms in the juncture node zone and lower arch at 2.8*P*; the longitudinal stiffening rib in the juncture zone at 3.4*P*; the webs in the lower arch at 3.6*P*; and the longitudinal stiffening rib in the lower arch at 4.3*P*. As a result, the model attains its ultimate capacity at 4.339*P*. Before the model failure, there is no visible curved phenomenon and little plastic strain in the longitudinal stiffening rib. Compared with the initial model, only the diaphragm reaches the plastic phase in advance, whereas the strains in other parts remain unchanged.

When the diaphragm thickness of the initial model is changed from 6 mm to 8 mm, the webs in the juncture node zone reach the plastic phase at 2.8*P*. Then, the sequence of plasticity initiation is ordered as follows: The longitudinal stiffening rib in the juncture node zone at 3.2*P*; the webs and stiffener in the lower arch at 3.6*P*; the diaphragms in the lower arch and node zone at 4.0*P*, whereas the model reaches its ultimate capacity at 5.0*P*. Before the model failure, no curved phenomenon and only little plastic strain were detected in the longitudinal stiffening rib. Compared with the initial model, only the diaphragm reached the plastic phase, whereas the other parts remain unperturbed.

The effects of the diaphragm on the structural behavior can be described in two aspects in accordance with the thickness changes:

1. Deformation

Load-displacement curves at several points of the model are depicted in Figure 16. In order to provide a sound comparison with the stiffener effects, the points are the same ones examined in Section 4.1.

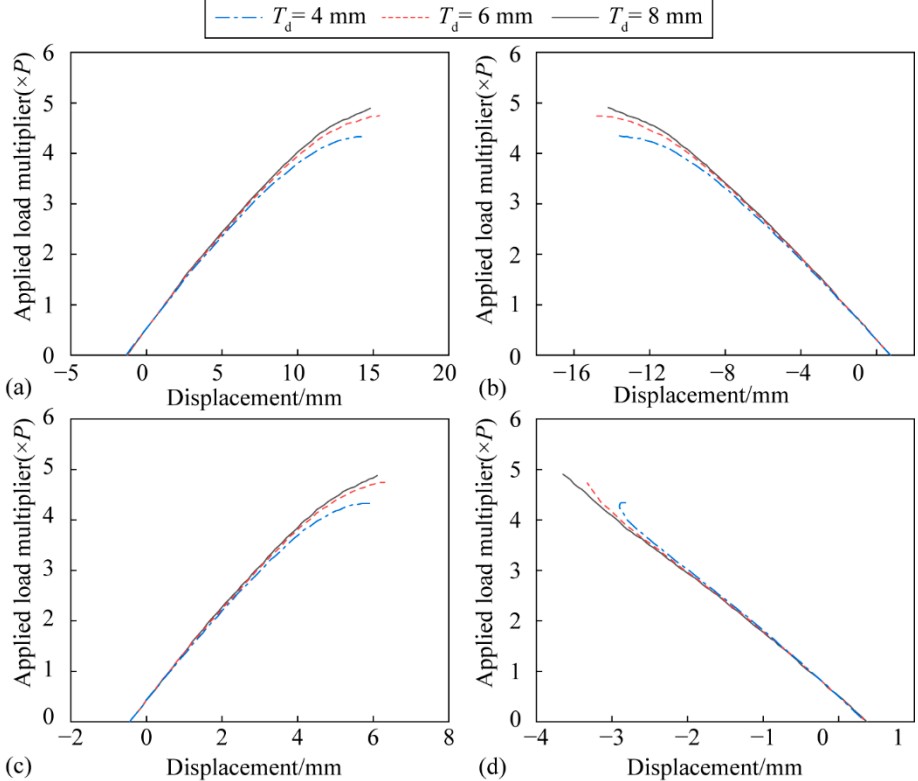

**Figure 16.** (**a**) Vertical displacement on the top flange, (**b**) horizontal displacement on the front web of the point in the juncture node zone, (**c**) vertical displacement on the top flange, (**d**) horizontal displacement on the front web of the point in the curved zone due to the change in the thickness of the diaphragm.

It can be generally seen that all relationships for different thicknesses are practically the same. This result illustrates that the diaphragm thickness scarcely contributes to the

structural deformation. The failure mode of the proposed model is observed as an overall elastic-plastic buckling.

2.    Plastic strain and ultimate capacity

It is noted that the ultimate capacity increases linearly with $T_d$. Such a trend occurs also in the maximal plastic strain in the stiffener and web, in contrast to that in the diaphragm. Furthermore, it is noted that the increasing of the diaphragm thickness does not change the failure mode of the model.

*4.3. Thickness of Web $T_w$*

To investigate the effects of web thickness, $T_w$ = 4 and 8 mm were selected and compared to that of the initial model with $T_w$ = 6 mm.

The plastic strain distribution contour plots in the web and stiffener are shown in Figure 17. When the web thickness of the initial model is changed from 6 mm to 4 mm, the web in the juncture node zone reaches the plastic phase at 2.0*P* first. The sequence of plasticity attainment in other parts is: the web in the lower arch at 2.4*P*; the diaphragm in the juncture node zone and lower arch at 2.8*P*; the longitudinal stiffening rib in the juncture node zone and lower arch at 3.0*P*, whereas the model reaches its ultimate capacity at 3.31*P*. Below 3.1*P*, there is no curving deformation in the web. There is an obvious curved phenomenon in the structure when the ultimate capacity is reached.

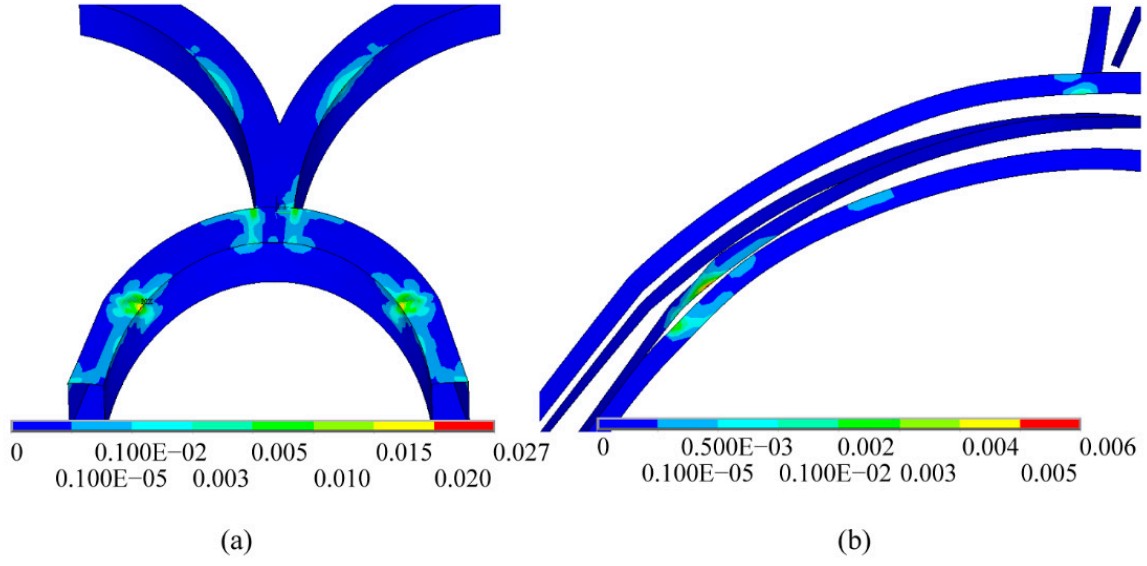

(a)                                                                                (b)

**Figure 17.** Plastic strain contour plot in the (**a**) web and (**b**) stiffener with $T_w$ = 4 mm.

When the web thickness of the initial model is changed from 6 mm to 8 mm, the web in the juncture node zone achieves the plastic phase at 3.4*P* first. The order of plasticity initialization in other parts is arranged as follows—the diaphragm in the juncture node zone at 3.6*P*, the diaphragm in the lower arch at 3.8*P*, the longitudinal stiffening rib in the juncture node zone at 4.0*P*, and the web and stiffener in the lower arch at 4.4*P*. The model reaches its ultimate capacity at 5.961*P*. Before failure, there is no visible curved phenomenon and little plastic strain in the stiffener. Hence, the effects of the web on the structural behavior can be summarized in three aspects:

1.    Deformation

The load-displacement curves of the typical points were selected to study the web thickness effects on deformation, as shown in Figure 18. In order to compare them with the influences of the stiffener and diaphragm, the points were the same ones considered previously.

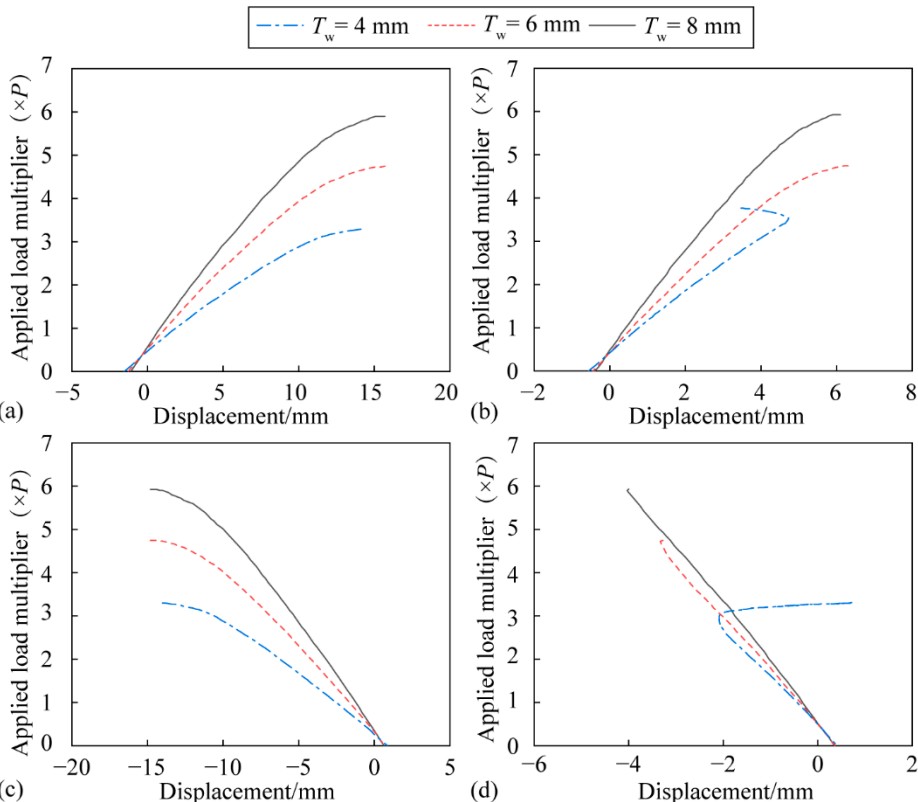

**Figure 18.** (**a**) Vertical displacement on the top flange, (**b**) horizontal displacement on the front web of the point in the juncture node zone, (**c**) vertical displacement on the top flange, (**d**) horizontal displacement on the front web of the point in the curved zone due to the change in the thickness of the web.

According to Figure 18a,b, the displacement in the juncture node zone and the slopes of the three curves decrease visibly in an opposing trend if compared with the increment of the web thickness. The slopes are near to zero when the load is close to the ultimate capacity. The slopes for the load-displacement relationship of the curved zone decrease in different levels with the web thickness enhancement, as displayed in Figure 18c,d. Thinner sections cause changes in the displacement and curved or dented phenomenon in the plate of the lower arch, resulting in lower ultimate capacity.

2.    Failure mode

In addition to the overall elastic-plastic buckling in the initial model with $T_w$ = 6 mm, if the web thickness is sufficiently low, the failure mode includes two elastic-plastic buckling modes: local and overall buckling.

3.    Plastic strain and ultimate capacity

The ultimate capacity increases linearly with the thickness (not shown). When the web is thinner, the plastic zone is concentrated on the lower arch. On the contrary, if the web is thicker, the plastic zone is concentrated on the juncture node zone, whereas the plastic strain in the zone below the node visibly decreases.

## 5. Conclusions

The influence of the parametric changes of initially curved and twisted box sections on the structural responses of the partial part of a dome, especially on the ultimate capacity, has been numerically investigated. Changes include the longitudinal stiffening rib, the diaphragm, and the web thicknesses. The stiffener thickness is a critical factor for the ultimate capacity. Omitting the stiffener causes the ultimate capacity to decrease, with a

larger curved zone in the lower arch. If the thickness of the stiffener is too small, it deforms prematurely and yields to premature failure. If the stiffener thickness is too large, the ultimate capacity will not be effectively improved. In general, the stiffener is useful to reduce the M11 moment in the plane formed by the axis and normal of the member plate. It also can reduce the M22 moment in the plane formed by the tangent and normal of the member plate. Nevertheless, the stiffener does not affect the overall rigidity of the model.

The ultimate capacity changes linearly with the change in the diaphragm thickness. The enhancement in thickness leads to increases in the maximal strain of the stiffener and web, while reducing the maximal strain of the diaphragm. The thickness increase can only delay the plastic phase of the diaphragm but does not change the failure mode of the structure. In addition, the diaphragm does not affect the overall rigidity of the model. The ultimate capacity of the model changes linearly with the change in the web thickness. A thinner web leads to the concentration of the plastic zone in the lower arch. When the web is thickened, the plastic zone is concentrated in the juncture node zone. In accordance with the load-displacement curve, the curve slope increases with the thickness. This proves that the increasing of the web thickness can visibly improve the overall rigidity of the proposed model.

The failure mode of the initial model is an overall elastic-plastic buckling. A thinner stiffener or web induces a visible curved zone in the lower arch under the ultimate capacity. In this case, the failure mode includes both local and overall buckling. The diaphragm thickness has little effect on the failure mode of the proposed model.

**Author Contributions:** Conceptualization, S.W. and J.C.; methodology, S.W. and J.F.; software, Z.W.; validation, C.P. and X.W.; investigation, H.W.; resources, J.F. and J.C.; data curation, Z.W. and C.P.; writing—original draft preparation, S.W. and Z.W.; writing—review and editing, X.W. and H.W.; visualization, Z.W.; supervision, J.F. and J.C.; funding acquisition, S.W. All authors have read and agreed to the published version of the manuscript.

**Funding:** This research was funded by the Natural Science Foundation of Gansu Province, grant number 20JR5RA079.

**Institutional Review Board Statement:** Not applicable.

**Informed Consent Statement:** Not applicable.

**Data Availability Statement:** All data needed to evaluate the conclusions in the paper are present in the paper. Additional data related to this paper may be requested from the corresponding author.

**Conflicts of Interest:** The authors declare no conflict of interest.

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
