# Peer review of "Structural Performance of Thin-Walled Twisted Box-Section Structure"

_buildings, doi:10.3390/buildings12010012_

Round 1

Reviewer 1 Report

Article is missing its authors and their associations at the start of the article.

Article is missing “Author Contributions”, “Funding”, “Acknowledgments” and “Conflicts of Interest” sections at the end of the article before the References.

The word “experiment” cannot be a keyword. It is not descriptive or identifying enough to fit the criteria.

Line 23: What qualifies as “good” rigidity? Is there a metric used to determine it? Identify it.

Line 113: The equation should be separate like Figures/Tables and should have an appropriate caption.

Line 122: “the lateralc view” should be “the lateral view”.

Line 380: “the webs in the juncture node zone reaches” should be either “the web in the juncture node zone reaches” or “the webs in the juncture node zone reach”.

Line 389: The same correction is needed as in line 380.

Reviewer 2 Report

Comments and suggestions for authors are contained in the attached document. 

Round 2

Reviewer 2 Report

I accept corrected paper.